# Predictors of antenatal care services utilization by pregnant women in Afghanistan: Evidence from the Afghanistan Health Survey 2018

**Muhammad Haroon Stanikzai**[1,2,3], **Essa Tawfiq**[4], **Charuai Suwanbamrung**[1,2], **Abdul Wahed Wasiq**[5], **Paleeratana Wongrith**[2,6] *

1 Public Health Research Program, School of Public Health, Walailak University, Tha Sala, Thailand,
2 Excellent Center for Dengue and Community Public Health (EC for DACH), Walailak University, Tha Sala, Thailand, 3 Department of Public Health, Faculty of Medicine, Kandahar University, Kandahar, Afghanistan, 4 The Kirby Institute, UNSW Sydney, Sydney, Australia, 5 Department of Internal Medicine, Faculty of Medicine, Kandahar University, Kandahar, Afghanistan, 6 School of Public Health, Walailak University, Tha Sala, Thailand

* paleerut@gmail.com

## Abstract

### Background

Poor utilization of maternal healthcare services remains a public health concern in most low and middle-income countries (LMICs), including Afghanistan. Late, inadequate, or no utilization of antenatal care (ANC) services pose a great concern.

### Objectives

This study assessed the predictors of ANC service utilization among Afghan pregnant women, using secondary data from the Afghanistan Health Survey 2018 (AHS2018).

### Methods

In this study, we used data from 10,855 ever-married women, aged 13–49 years, who gave birth in the two years prior to the survey or those women who were currently pregnant. The outcome variable was defined as a binary variable on ANC utilization (≥1 ANC visit equals 1, and zero otherwise). We fitted a binary logistic regression model and examined the associations between ANC utilization and explanatory variables, providing findings based on univariate and multivariate analysis. STATA version 17 was employed for the data analysis.

### Results

Overall, 63.2%, 22.0%, and 3.1% of women had at least one ANC visit, ≥4 ANC visits, and ≥8 ANC visits during their last pregnancy, respectively. Higher odds of ANC utilization were observed in women who could read and write (AOR = 1.55, 95%CI: 1.36–1.77), whose husbands could read and write (AOR = 1.42, 95%CI: 1.29–1.56), who knew 1 sign (AOR = 1.93, 95%CI: 1.74–2.14), 2 signs (AOR = 2.43, 95%CI: 2.17–2.71) and ≥ 3 signs (AOR = 1.55,

**Data Availability Statement:** Data that support the findings of the current study are stored in the Monitoring and Evaluation Department at the

Ministry of Public Health (MoPH), Afghanistan. Data access may be granted upon approval from the data custodians at MoPH, Afghanistan (Email: info.access@moph.gov.af or atasayedzai@gmail. com). Concerns raised about the data analysis and findings in the present study can also be brought to the attention of the Research and Ethics Committee, Faculty of Medicine, Kandahar University (Email:najeebrahimy@gmail.com).

**Funding:** This study was financially supported by the School of Public Health and Excellent Center for Dengue and Community Public Health, Walailak University. The funders had no role in study design, data collection and analysis, decision to publish, or preparation of the manuscript.

**Competing interests:** The authors have declared that no competing interests exist.

95%CI: 1.36–1.77) of complicated pregnancy, and those with almost daily access to radio (AOR = 1.19, 95%CI: 1.08–1.327) and TV (AOR = 1.92, 95%CI: 1.73–2.13). We also found that women with one (AOR = 0.64, 95%CI: 0.49–0.84) and more than one (AOR = 0.60, 95%CI: 0.47–0.76) parity status, those for whom in-laws and others decided for their birth-place [(AOR = 0.85, 95%CI: 0.74–0.97) and (AOR = 0.63, 95%CI: 0.55–0.72), respectively], and those that resided in rural areas (AOR = 0.89, 95%CI: 0.79–1.00) had reduced odds of ANC utilization.

## Conclusion

ANC service utilization is unacceptably low by pregnant women in Afghanistan. The predictors of ANC utilization identified by the study should be considered in the design of future interventions to enhance antenatal care utilization in Afghanistan.

## Introduction

Maternal mortality has been steadily decreasing worldwide, from more than 380 deaths to an estimated 211 deaths per 100,000 live births, between the 1900s and 2020 [1]. However, the progress is highly unequal, owing to stark geographic disparities [1, 2]. Reductions in maternal mortality have been comparatively modest in low and middle-income countries (LMICs), where 94% of the global maternal deaths occur [3, 4]. In addition, epidemiologic studies suggest that maternal mortality and morbidity appear to have a measurable impact on newborn health [5–9], emphasizing the need for comprehensive maternity care during pregnancy in these settings.

Antenatal care (ANC), a core intervention of the safe motherhood initiatives, anticipates improved maternal and neonatal outcomes [10–12]. Research indicates that the use of high-quality ANC services, characterized by timely and frequent visits, can potentially reduce pregnancy-related maternal deaths by 20% [8]. Moreover, the benefits of a single ANC visit have been widely documented in the literature [13–18]. Aside from the obvious health benefits, most of the contents in ANC services bear minimal costs to both clients and providers [19, 20]. For example, a systematic review across LMICs revealed that ANC utilization tends to lower costs for uncomplicated services during pregnancy [21]. Furthermore, a study conducted in Eastern Sub-Saharan Africa and South-East Asia details the opportunities for cost savings in ANC services [22]. Taken together, these data suggest that antenatal care could be one of the cost-effective strategies to improve maternal, newborn, and child health (MNCH) in LMICs.

Despite the overwhelming evidence that antenatal care visits are associated with significant health and economic benefits, millions of pregnant women, predominantly in LMICs, do not attend health facilities for ANC services. More specifically, 10–15% of pregnant women do not receive at least one ANC visit during pregnancy in these settings [23–26]. Furthermore, an alarming proportion of pregnant women do not utilize a single ANC visit as reflected in population-based surveys across LMICs [15, 16, 27–29]. According to the Afghanistan Demographic Health Survey 2015 (DHS), 58.6% of pregnant women complete only a single ANC visit during pregnancy [30, 31], which is a clear reflection of poor maternal healthcare utilization.

A number of local studies have described the attributes of ANC utilization among Afghan pregnant women in recent years, including determinants of utilization [30, 32], timing and

frequency of ANC visits [33–35], and contents of these services [36]. However, very few studies focused on pregnant women not utilizing a single ANC visit, thus a complete picture of the ANC utilization is lacking on a national level, given that a single ANC visit is associated with well-documented health benefits and an opportunity to encourage pregnant women to adhere to high-quality ANC services [10]. Moreover, at least one ANC visit is one of the three indicators used to track the progress of ANC services utilization at the population level across different countries [37–39].

To address this gap in the literature from Afghanistan, we used data from the Afghanistan Health Survey 2018 (AHS2018) with the objective to determine the prevalence of ANC services utilization and its predictors in Afghanistan. Documenting the attributes of ANC coverage on a national level could inform health policy and planning for maternal health.

## Methods

We used data from the Afghanistan Health Survey 2018 (AHS 2018) and we accessed the dataset on 13/12/2023. AHS 2018 employed a stratified two-stage sampling design. Details of the sampling approach are available elsewhere [40]. During the survey, women were interviewed by trained surveyors, using a questionnaire on seeking healthcare for women and children.

In this study, we used data from ever-married women, aged 13–49 years, who gave birth in the two years prior to the survey or those women who were currently pregnant. The latest pregnancy either in the past two years or the current pregnancy was used to minimize recall bias by the respondent women. A total of 10,855 women met our inclusion criteria (above), and their data were analyzed. The survey questionnaire was designed to collect data from ever-married women aged 12–49 years. However, the AHS 2018 data used in our study did not have data from ever-married women aged 12 years.

### Study variables

The outcome variable was defined as a binary variable on ANC utilization ($\geq$1 ANC visit equals 1, and zero otherwise).

The decision on which explanatory variables need to be included in the multivariate analysis was made by the relevance and potential effects of the variables on the outcome of interest (ANC utilization), based on the evidence reported in recent studies from Afghanistan [33, 36]. The explanatory variables included in the analysis were: woman's age (13–29 years, 30–39 years, and 40–49 years), woman's literacy (whether or not she could read and write), husband's literacy (whether or not he could read and write), parity (the woman had not given birth [nullipara], the woman had given birth once (primipara), the woman had given birth at least twice [multipara]), knowledge on the number of danger signs in pregnancy (no knowledge, knows 1 sign, knows 2 signs, knows $\geq$3 signs), residential area (urban vs rural), household size (2–4 people vs $\geq$5 people), decision made for woman where to give birth (by herself, her husband, in-laws, others), access to the internet (whether or not she used the internet almost daily for consulting for her healthcare), access to the radio (whether or not she listened to the radio almost daily), and access to TV (whether or not she watched TV almost daily).

### Statistical analysis

We conducted a descriptive analysis of the baseline characteristics of women (e.g., age, education) and other explanatory variables included in the multivariate analysis. We also examined the distribution of ANC utilization, such as proportions of women with no ANC visit, 1 ANC visit, 2 ANC visits up to $\geq$ 8 ANC visits, and proportions of women with ANC utilization by urban vs. rural areas, by timeliness (early vs. late) of ANC visits, as well as reasons for not

using ANC services in women who did not use services in their latest pregnancy. The reasons for not using ANC services were collected as part of the structured response options in the survey questionnaire.

We fitted a binary logistic regression model and examined the associations between ANC utilization and explanatory variables, providing univariate and multivariate results. Only variables with *p* value of less than 0.25 in the bivariate analysis were included for multivariate logistic regression analysis. To take the clustering effects of data at the household level into account, we added a random cluster effect in our model estimates, and we adjusted standard errors for the odds ratios (ORs) and 95% CIs. STATA version 17 was employed for the data analysis.

## Ethical approval

The study was reviewed by the Research and Ethics Committee, Faculty of Medicine, University of Kandahar, Afghanistan (Certificate #98; Dated 14/November/2023). The committee approved the study and waived the ethical application because secondary data from the Afghanistan Health Survey 2018 (AHS2018) were used and analysed in this study. For the AHS 2018, ethical approval was obtained from the Institutional Review Board of the Ministry of Public Health of Afghanistan in 2017. A written informed consent to proceed with the interviews was obtained from all participants. For pregnant women aged less than 16 years, written informed consent was obtained from their legal guardians (husbands/fathers-in-law/mothers-in-law).

## Results

Table 1 presents baseline characteristics of 10,855 ever-married women by status of ANC services utilization during their latest pregnancies. Nearly two-thirds (62.7%) of women were 13–29 years old. The proportion of women and proportion of husbands with primary and secondary/higher education were higher for women who utilized ANC services than those who did not, and the differences were statistically significant. In terms of parity, the differences between the two groups of women were not statistically significant. Overall, 43.2% of women did not know any sign of risk during pregnancy; however, there were significant differences between the two groups of women, with women who utilized ANC services knowing more signs of complicated pregnancy. In terms of household size, there were no statistically significant differences between the two groups of women. Overall, most women (77.5%) lived in rural areas, and there were significant differences between the two groups of women, with higher ANC utilization in urban areas and lower ANC utilization in rural areas. There were significant differences between women who utilized ANC services and those who did not in women who themselves decided (29.9% vs. 26.7%), and whose husbands decided (38.1% vs. 36.1%) for birth-place choice. The proportions of women with almost daily access to the internet, radio, and TV were significantly different between the two groups of women, with higher ANC utilization by women with access to the media than women with no daily access to the media.

Fig 1 presents ANC utilization by pregnant women during their latest pregnancies. It shows that 1 out of 3 pregnant women did not have any ANC visits, and nearly two-thirds of women attended at least one ANC visit, although only 3.1% attended ≥8 ANC visits. Additionally, it shows that a higher proportion of pregnant women did not attend ANC visits in rural areas compared to urban areas.

Fig 2 depicts ANC utilization by timeliness of ANC commencement—timely vs. late ANC. It shows that a higher proportion of women with 1 ANC visit commenced their ANC visit during the first trimester. However, the opposite was observed in women with ≥3 ANC visits,

**Table 1. Baseline characteristics of pregnant women by status of ANC services utilization.**

| | | Total | Did not use ANC services | Used ANC services | p-value |
|---|---|---|---|---|---|
| | | n = 10,855 (100%) | n = 3,992 (36.8%) | n = 6,863 (63.2%) | |
| Woman's age | 13–29 years | 62.7% | 62.2% | 63.1% | 0.22 |
| | 30–39 years | 30.3% | 30.3% | 30.2% | |
| | 40–49 year | 7.0% | 7.5% | 6.7% | |
| Woman's education | No formal education | 94.5% | 96.1% | 93.7% | < 0.001 |
| | Primary education | 3.5% | 2.8% | 3.8% | |
| | Secondary/higher education | 2.0% | 1.1% | 2.5% | |
| Husband's education | No formal education | 98.2% | 98.6% | 97.9% | 0.03 |
| | Primary education | 0.2% | 0.2% | 0.3% | |
| | Secondary/higher education | 1.6% | 1.2% | 1.8% | |
| Parity | Nulliparous | 3.9% | 3.6% | 4.1% | 0.23 |
| | Primiparous | 12.7% | 12.3% | 12.9% | |
| | Multipara | 83.4% | 84.1% | 83.0% | |
| Knowledge of the number of danger signs | None | 43.2% | 57.4% | 35.0% | < 0.001 |
| | 1 | 25.2% | 21.7% | 27.3% | |
| | 2 | 22.8% | 16.1% | 26.6% | |
| | ≥3 | 8.8% | 4.8% | 11.1% | |
| Household size | 2–4 persons | 13.4% | 13.8% | 13.2% | 0.18 |
| | ≥5 persons | 86.6% | 86.2% | 86.8% | |
| Residential area | Urban | 22.5% | 16.7% | 25.8% | < 0.001 |
| | Rural | 77.5% | 83.3% | 74.2% | |
| Decision made for woman where to give birth | Herself | 28.7% | 26.7% | 29.9% | < 0.001 |
| | Husband | 37.3% | 36.1% | 38.1% | |
| | In-laws | 15.9% | 16.4% | 15.5% | |
| | Others | 18.1% | 20.8% | 16.5% | |
| Access to internet | No access | 97.3% | 98.5% | 96.6% | < 0.001 |
| | Almost daily access | 2.7% | 1.5% | 3.4% | |
| Access to radio | No access | 75.7% | 79.6% | 73.4% | < 0.001 |
| | Almost daily access | 24.3% | 20.4% | 26.6% | |
| Access to TV | No access | 67.9% | 80.0% | 60.9% | < 0.001 |
| | Almost daily access | 32.1% | 20.0% | 39.1% | |

The chi square test was used to examine the level of statistical significance and to report p-values

with an increasing gap between women with timely and late commencement of ANC visits by the number of ANC visits.

Fig 3 presents the reasons pregnant women provided for not using ANC services. Of 7,306 responses, provided by the 3,992 women who did not use ANC services, 30% of reasons was "it was not necessary to use ANC services". Other reasons were—health facilities too far or no transport to health facilities (18%), too expensive transportation cost (14%), security concerns (11%), too expensive services (9%), unfriendly staff (5%), religious reasons (5%), inconvenient service hours (4%), not having an accompanying man—mahram (2%), no female staff (2%), and other reasons (0.1%).

Table 2 presents odds ratios (ORs) on the likelihood of ANC utilization by ever-married women during their latest pregnancies. Results from the multivariable analysis show that it was more likely that women who could read and write attend ANC visits than women who could not [AOR 1.55 (95% CI: 1.36–1.77)]. Husband's literacy was associated with ANC

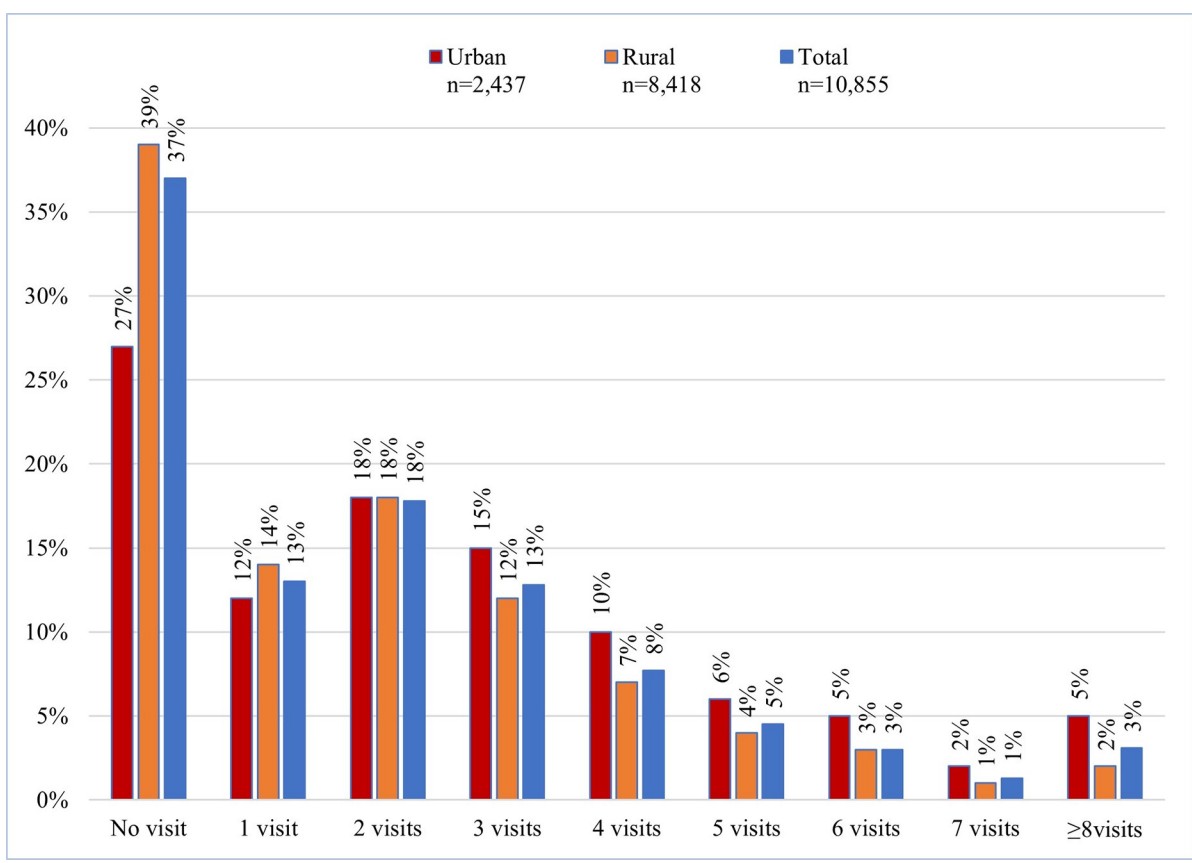

**Fig 1. Utilization of ANC services by pregnant women (n = 10,855).**

utilization, with AOR 1.42 (95% CI: 1.29–1.56) for husbands who could read and write, compared to husbands who could not. In terms of parity, women of primiparous and women of multiparous were less likely to utilize ANC, compared to women of nulliparous but currently pregnant [AOR 0.64 (95% CI: 0.49–0.84), and 0.60 (0.47–0.76), respectively]. Women with knowledge of danger sign(s) during pregnancy were more likely to utilize ANC services [AOR 1.93 (95% CI: 1.74–2.14) for those who knew one sign, 2.43 (2.17–2.71) for those who knew 2 signs, and 3.17 (2.66–3.77) for those who knew ≥3 signs, compared to women who knew no signs]. Geographic location was a strong predictor of ANC utilization, with AOR (95% CI) of 0.89 (0.79–0.93) for women who lived in rural areas, compared to women who lived in urban areas. Decisions made for women where to give birth was a strong predictor of ANC utilization [AOR 0.85 (95% CI: 0.74–0.97) for the decision made by in-laws, and 0.63 (0.55–0.72) for the decision made by others for the women, compared to the decision made by women themselves]. Access to mass media was another significant predictor of ANC utilization [AOR 1.19 (95% CI: 1.08–1.32) for women with almost daily access to radio, and 1.92 (1.73–2.13) for women with almost daily access to TV, compared to women with no daily access to radio, and TV, respectively].

## Discussion

Our analysis of the ANC services utilization among pregnant women in Afghanistan showed concerning results. There were 63.2% of pregnant women that received at least one ANC visit,

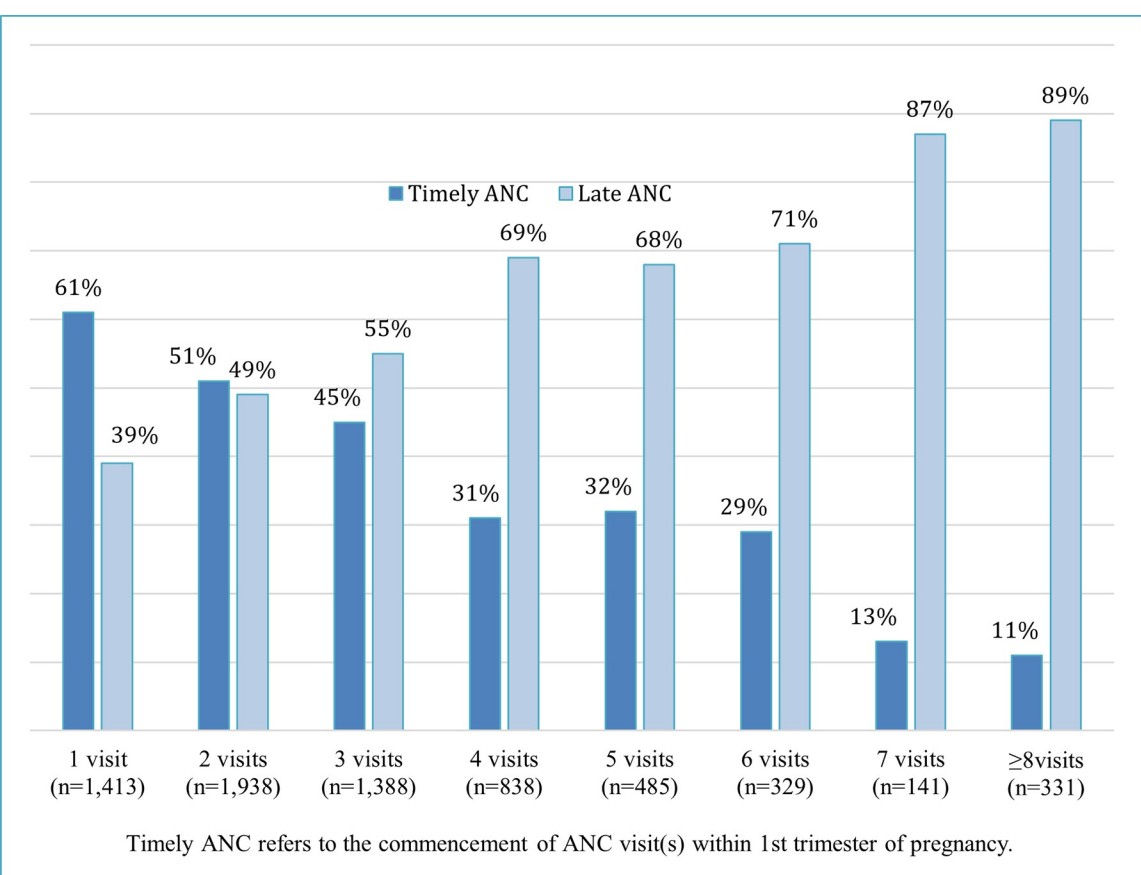

**Fig 2. ANC service utilization by timeliness of ANC commencement—Timely vs. Late ANC.**

with 22% of pregnant women receiving ≥ 4 ANC visits and only 3.1% ≥ 8 ANC visits. Many factors were found to be potentially related to ANC service utilization, including woman and husband education, parity, knowledge of pregnancy danger signs, residential location, influence of decision-makers on place of delivery, and media access.

In this study, the proportion (63.2%) of pregnant women with at least one ANC visit revealed a slight increase from the findings reported in the national health surveys undertaken in 2012 (54%) and 2015 (58.6%) [30, 31]. However, this proportion is lower than those reported from other LMICs, including 85% in Somalia [41], 93.8% in India [37], 74.3% in Ethiopia [42], 92.1% in Nepal [43], and 92.5% in Pakistan [38]. Moreover, the increase reported over time of ≥ 4 ANC visits and ≥ 8 ANC visits in Afghanistan is much slower than those achieved in other LMICs [37, 38, 42, 43]. Therefore, increased efforts are needed to reach pregnant women who do not utilize ANC services and move forward toward achieving the WHO-recommended ANC services [10].

The most common reasons for not utilizing ANC services were low perceived needs for ANC services, transportation barriers, security concerns, financial constraints, and unfriendly behavior of healthcare workers. Similar reasons were reported by Stanikzai and colleagues in a study on the determinants of antenatal care utilization in Kandahar, Afghanistan [32]. Immediate policy efforts and tailored interventions are needed to tackle the mentioned barriers to ANC services utilization.

The present study identified a strong association between women's educational status and ANC service utilization. As meta-analyses have highlighted, a woman's educational attainment

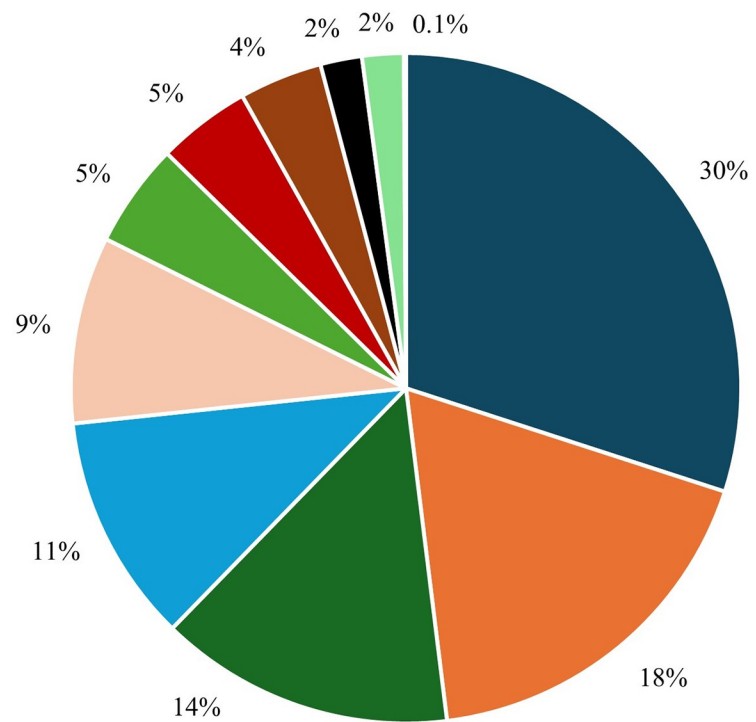

- It was not necessary (30%)
- Too far, no transportation (18%)
- Too expensive transportation cost (14%)
- Security concerns (11%)
- Services too expensive (9%)
- Unfriendly staff (5%)
- Religious belief (5%)
- Inconvenient service hours (4%)
- No man to accompany (2%)
- No female provider at facility (2%)
- Other reasons (0.1%)

**Fig 3. Reasons provided by 3,992 women who did not use ANC services (n = 7,306 responses).**

is a strong predictor of ANC service utilization [39, 44]. These findings merit greater emphasis in women's education on healthcare utilization during pregnancy [38, 45, 46]. Unfortunately, the recent restrictions on female education in Afghanistan are concerning [47, 48]. Therefore, our findings warrant an urgent focus on female education and suitable public health policies to increase awareness among uneducated pregnant women about the consequences of not using ANC services.

Consistent with previous studies from Afghanistan [33, 36, 49], we found that women whose husbands could read and write were more likely to receive ANC services than those whose husbands could not. This finding is echoed in previous observations regarding the influence of husbands' education on ANC service utilization across LMICs [39, 50]. Given the current sociopolitical challenges in the country, engaging husbands in reproductive health interventions, may also be a key intervention [51].

The lower likelihood of ANC service utilization was reported in primipara and multipara pregnant women in our study. This is in line with studies done in other LMICs [52, 53]. Additionally, higher parity is reported as a barrier to receiving timely [54–56] and frequent ANC visits across different studies [56, 57]. Based on the findings of these studies, women who had a previous pregnancy are less motivated to receive ANC services, assuming that they may have already possessed sufficient knowledge and experience from prior pregnancies and childbirths. Nevertheless, in a systematic review, the association between parity and ANC utilization was not ascertained [39]. Interventions for sensitizing primipara and multipara pregnant women may be useful to improve the utilization of ANC services for these groups of pregnant women.

We found statistically significant associations between women's knowledge of complicated pregnancy and higher odds of ANC service utilization. The effects of women's knowledge of obstetric danger signs on their health-seeking behaviors are encouraging, as documented in

**Table 2. Likelihood of ANC services utilization by pregnant women.**

| | COR (95%CI) n = 10,855 | p-value | AOR (95%CI) n = 10,855 | p-value |
|---|---|---|---|---|
| Woman's age | | | | |
| 13–29 years | Ref | | - | |
| 30–39 years | 0.99 (0.90–1.07) | 0.22 | - | - |
| 40–49 year | 0.88 (0.75–1.03) | 0.11 | - | - |
| Woman can read and write | | | | |
| No | Ref | | Ref | |
| Yes | 2.46 (2.19–2.77) | < 0.001 | 1.55 (1.36–1.77) | < 0.001 |
| Husband can read and write | | | | |
| No | Ref | | Ref | |
| Yes | 1.85 (1.69–2.01) | < 0.001 | 1.42 (1.29–1.56) | < 0.001 |
| Parity | | | | |
| Nulliparous | Ref | | Ref | |
| Primiparous | 0.92 (0.73–1.15) | 0.18 | 0.64 (0.49–0.84) | < 0.001 |
| Multipara | 0.86 (0.70–1.06) | 0.15 | 0.60 (0.47–0.76) | < 0.001 |
| Knowledge of the number of danger signs | | | | |
| None | Ref | | Ref | |
| 1 | 2.07 (1.87–2.28) | < 0.001 | 1.93 (1.74–2.14) | < 0.001 |
| 2 | 2.71 (2.43–3.02) | < 0.001 | 2.43 (2.17–2.71) | < 0.001 |
| ≥3 | 3.75 (3.17–4.44) | < 0.001 | 3.17 (2.66–3.77) | < 0.001 |
| Household size | | | | |
| 2–4 persons | Ref | | - | |
| ≥5 persons | 1.05 (0.94–1.18) | 0.18 | - | - |
| Residential area | | | | |
| Urban | Ref | | Ref | |
| Rural | 0.58 (0.52–0.64) | < 0.001 | 0.89 (0.79–0.93) | 0.04 |
| Decision made for woman where to give birth | | | | |
| Herself | Ref | | Ref | |
| Husband | 0.94 (0.85–1.04) | 0.24 | 1.02 (0.91–1.13) | 0.78 |
| In-laws | 0.84 (0.75–0.96) | 0.01 | 0.85 (0.74–0.97) | 0.01 |
| Others | 0.71 (0.63–0.80) | < 0.001 | 0.63 (0.55–0.72) | < 0.001 |
| Access to internet | | | | |
| No access | Ref | | - | |
| Almost daily access | 2.23 (1.67–2.98) | < 0.001 | - | - |
| Access to radio | | | | |
| No access | Ref | | Ref | |
| Almost daily access | 1.41 (1.29–1.56) | < 0.001 | 1.19 (1.08–1.32) | < 0.001 |
| Access to TV | | | | |
| No access | Ref | | Ref | |
| Almost daily access | 2.57 (2.32–2.82) | < 0.001 | 1.92 (1.73–2.13) | < 0.001 |

Abbreviations: COR and AOR refer to crude and adjusted odds ratios, respectively.

this study and earlier literature [33, 58, 59]. A considerable proportion of pregnant women is likely to have poor knowledge of complicated pregnancy in LMICs [60, 61]. Likewise, nearly half (43.2%) of the pregnant women in this study were not aware of the signs associated with complicated pregnancy. Efforts should focus on developing interventions to increase awareness of complicated pregnancy in pregnant women.

Given the higher socioeconomic adversities in rural Afghanistan [62], the lower utilization of ANC services in these settings is not far-fetched. A growing body of literature has documented geographical disparities in ANC service utilization [26, 53, 57]. The difference in ANC service utilization between rural and urban areas in Afghanistan reflects broader healthcare gaps [63, 64], heightened insecurity [63, 65], and various socioeconomic challenges [66, 67]. Such conditions in Afghanistan's rural settings magnify the challenges of ANC utilization, underscoring the urgent need for targeted, multi-sectoral interventions to bridge these gaps and support the well-being of pregnant women in rural areas.

This study also adds to the existing literature that women's and husbands' decisions on birth place choice were strong predictors of ANC service utilization. Studies elsewhere reported that increasing women's autonomy and husbands' involvement in maternal healthcare enhances ANC service utilization [68–70]. However, the deeply ingrained sociocultural values in Afghanistan frequently restrict women and young couples from individual preference [71]. It is imperative to empower women with decision-making autonomy, ensuring they have the right and dignity to choose and access quality health services.

We found that access to media (radio and TV) was associated with higher ANC service utilization, and this is true across studies in Afghanistan [33, 36], and elsewhere [39, 72]. This suggests that mass media could influence pregnant women to utilize maternal healthcare by broadcasting maternal health service utilization educational programs in low-resource settings [73]. Our finding has important implications and calls on the Afghan government, international partners, and other health stakeholders to continue allocating resources for mass media-centric educational programs to promote maternal health service utilization in Afghanistan.

Additionally, this study identified that a significant proportion (39%) of pregnant women commenced their ANC lately. In this study, pregnant women with late ANC visits reported a higher frequency of ANC visits than those with early ANC visits. This finding is inconsistent with most studies in LMICs [23, 26, 55], and requires further work, especially given the high prevalence of late ANC initiation in the study population. The details of ANC initiation and ANC contents are described elsewhere [34, 36].

## Limitations

There were some limitations in our study. First, there is a risk of recall bias as women recalled events from several months ago; women may have under-reported the events and related attributes. Second, data on antenatal care utilization relied on self-report, which may not be free from bias (social desirability bias). Third, we measured the outcome variable in terms of attending ≥1 ANC visit while the WHO ANC guideline recommends attending ≥8 ANC visits [10]. Considering a similar approach in previous studies [29, 39, 41] and low ANC coverage in Afghanistan [30], at least a single ANC visit used as the outcome variable in this study was methodologically appropriate. Fourth, the collected information in AHS2018 confined our evaluation of diverse predictors for ANC utilization. Therefore, we urge future studies to consider other influencing factors, such as access to healthcare facilities, social support systems, and health service factors in their analyses. Fifth, AHS was conducted in 2018, and as a consequence of the current sociopolitical situation, decreased donor funding, and the impact of the COVID-19 pandemic, there is a possibility that healthcare services in the country may have experienced a substantial decline [74, 75]. As such, the present analysis should be interpreted with caution, acknowledging that the findings may not fully reflect the current realities faced by the Afghan health system. Finally, the cross-sectional nature of our study precludes causal relationships between the predictors and ANC service utilization.

Despite the above limitations, this study enriches the literature by spotlighting the intersection of ANC utilization with the sociodemographic adversities faced by pregnant women in Afghanistan, advocating for policy reforms and the establishment of urgent tailored interventions for maternal health services.

## Conclusion

This study indicates that ANC service utilization is unacceptably low by pregnant women in Afghanistan. Improvement in ANC service utilization should be a public health priority for the Afghan government, UN agencies, and healthcare stakeholders to reduce the burden of pregnancy-related morbidity and mortality in Afghanistan. The predictors identified in this study should be considered in the design of future interventions for enhancing ANC services utilization in Afghanistan.

## Acknowledgments

We extend our gratitude to the Ministry of Public Health of Afghanistan for providing access to the data from the Afghanistan Health Survey 2018.

## Author Contributions

**Conceptualization:** Muhammad Haroon Stanikzai, Essa Tawfiq, Charuai Suwanbamrung, Abdul Wahed Wasiq, Paleeratana Wongrith.

**Data curation:** Muhammad Haroon Stanikzai, Essa Tawfiq, Paleeratana Wongrith.

**Formal analysis:** Muhammad Haroon Stanikzai, Essa Tawfiq, Charuai Suwanbamrung.

**Funding acquisition:** Paleeratana Wongrith.

**Investigation:** Muhammad Haroon Stanikzai, Essa Tawfiq, Charuai Suwanbamrung.

**Methodology:** Muhammad Haroon Stanikzai, Essa Tawfiq, Charuai Suwanbamrung, Paleeratana Wongrith.

**Project administration:** Muhammad Haroon Stanikzai.

**Resources:** Muhammad Haroon Stanikzai, Paleeratana Wongrith.

**Software:** Muhammad Haroon Stanikzai, Essa Tawfiq, Charuai Suwanbamrung.

**Supervision:** Charuai Suwanbamrung, Abdul Wahed Wasiq, Paleeratana Wongrith.

**Validation:** Muhammad Haroon Stanikzai, Paleeratana Wongrith.

**Visualization:** Muhammad Haroon Stanikzai.

**Writing – original draft:** Muhammad Haroon Stanikzai, Essa Tawfiq, Charuai Suwanbamrung, Abdul Wahed Wasiq, Paleeratana Wongrith.

**Writing – review & editing:** Muhammad Haroon Stanikzai, Essa Tawfiq, Charuai Suwanbamrung, Abdul Wahed Wasiq, Paleeratana Wongrith.

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
