## [Decision Letter · Decision Letter 0]

10 Mar 2024

PONE-D-24-05385Predictors of Antenatal Care Services Utilization by Pregnant Women in Afghanistan: Evidence from the Afghanistan Health Survey 2018PLOS ONE

Dear Dr. Wongrith,

Thank you for submitting your manuscript to PLOS ONE. After careful consideration, we feel that it has merit but does not fully meet PLOS ONE’s publication criteria as it currently stands. Therefore, we invite you to submit a revised version of the manuscript that addresses the points raised during the review process. Please submit your revised manuscript by Apr 24 2024 11:59PM. If you will need more time than this to complete your revisions, please reply to this message or contact the journal office at plosone@plos.org. Please include the following items when submitting your revised manuscript:A rebuttal letter that responds to each point raised by the academic editor and reviewer(s). You should upload this letter as a separate file labeled 'Response to Reviewers'.A marked-up copy of your manuscript that highlights changes made to the original version. You should upload this as a separate file labeled 'Revised Manuscript with Track Changes'.An unmarked version of your revised paper without tracked changes. You should upload this as a separate file labeled 'Manuscript'.If applicable, we recommend that you deposit your laboratory protocols in protocols.io to enhance the reproducibility of your results. Protocols.io assigns your protocol its own identifier (DOI) so that it can be cited independently in the future. For instructions see: https://journals.plos.org/plosone/s/submission-guidelines#loc-laboratory-protocols. Additionally, PLOS ONE offers an option for publishing peer-reviewed Lab Protocol articles, which describe protocols hosted on protocols.io. Read more information on sharing protocols at https://plos.org/protocols?utm_medium=editorial-email&utm_source=authorletters&utm_campaign=protocols.

We look forward to receiving your revised manuscript.

Kind regards,

Zahra Hoodbhoy

Academic Editor

PLOS ONE

Journal Requirements:

2. In this instance it seems there may be acceptable restrictions in place that prevent the public sharing of your minimal data. However, in line with our goal of ensuring long-term data availability to all interested researchers, PLOS’ Data Policy states that authors cannot be the sole named individuals responsible for ensuring data access (http://journals.plos.org/plosone/s/data-availability#loc-acceptable-data-sharing-methods).

Reviewers' comments:

Reviewer's Responses to Questions

**Comments to the Author**

1. Is the manuscript technically sound, and do the data support the conclusions?

Reviewer #1: Yes

Reviewer #2: Yes

2. Has the statistical analysis been performed appropriately and rigorously? 

Reviewer #1: Yes

Reviewer #2: Yes

3. Have the authors made all data underlying the findings in their manuscript fully available?

Reviewer #1: Yes

Reviewer #2: Yes

4. Is the manuscript presented in an intelligible fashion and written in standard English?

Reviewer #1: No

Reviewer #2: Yes

5. Review Comments to the Author

Reviewer #1: Abstract:

1. Conclusion: L50 talks about the improvement, however the earlier parts don't talk about assessing this

Introduction:

1. L73-6: How are ANC cost effective in current settings? need elaboration.

2. L83 and L86 already answers the objective of the study in Introduction.

Methods:

1. Methods described aren't the ones opted by the author They were done part of the National Survey. These methods should be supported be relevant citations.

2. Should also talk about the reasons that were found from women on not opting for ANC. Was that a qualitative part of the study? Or reasons were asked and later coded?

Results:

1. Table 1 should have p values in a separate column, especially when the result section talks so much about the SIGNIFICANT differences among the variables for the ANC and No ANC utilisation. The Asterisks used aren't very clear because few variables don't have any asterisks.

2. There is a lot of repetition in the text and tables.

3. Table 2 can highlight the factors which came significant in final adjusted model so it's easy for reader.

Discussion:

1. Authors can talk more about the modifiable factors that can increase the ANC visits, many factors identified by women who didn't went for any ANC visits should be worked on.

Limitations:

1. Please cite where required. like WHO guidelines L362, please recheck if its >8 or 4 ANC visits.

Reviewer #2: I would like to acknowledge the authors of ‘Predictors of Antenatal Care Services Utilization by Pregnant Women in Afghanistan: Evidence from the Afghanistan Health Survey 2018’ for all the hard work on this relevant topic and although the data used from 2018 might not portray the true picture in current scenario but nonetheless would help raise a highly pertinent issue.

The authors have done a good job in the analysis and writeup. I have a few suggestions below:

Methods

The data analysis part would need more details and also what cutoff was used at the univariate stage to include variables for MV analysis or were all variables included in the MV analysis. Ideally there is a cut-off for variables to be included in the MV.

It would have been good if the authors would also have looked at the factors for early vs. late ANC and if they vary by age on women. I understand this would take time but a suggestion to make this paper more policy relevant.

Results

The length of the results section needs to be shortened especially the description in text for table 1 and fig 1 and 2. As most of the information is there in the table and figures, the text should only focus on the key findings rather describing all.

The language and the flow of the writing can be improved especially when describing the MV analysis results, like this is not clear ‘Women’s and husbands’ decision on birth-place choice was a strong predictor of ANC utilization [OR 0.85 (95% CI: 0.74-0.97) for decision made by in laws, and 0.63 (0.55-0.72) for decision made by others for the women, compared to the decision made by women themselves].’ The flow and phrasing needs to be clear.

Discussion

This section can be made more succinct as it is currently lengthy.

Also discuss reasons of why women having late ANC had greater number of ANC visits

Figures

Fig 1 can be omitted as fig 2 has the required information.

6. PLOS authors have the option to publish the peer review history of their article (what does this mean?). If published, this will include your full peer review and any attached files.

Reviewer #1: No

Reviewer #2: No

---

## [Author Response · Author response to Decision Letter 0]

17 Mar 2024

Dear Editor,

We would like to thank the respected editor and the respected reviewers for their thoughtful evaluation of our manuscript (ID Number, PONE-D-24-05385) entitled “Predictors of Antenatal Care Services Utilization by Pregnant Women in Afghanistan: Evidence from the Afghanistan Health Survey 2018”. Please find our revised manuscript (with highlighted changes), which we believe is substantially strengthened now that we have incorporated reviewers’ recommendations. 

Journal requirements

Response: Thank you. We have made the manuscript including tables, figures, and references as per the journal’s style requirements.

2. In this instance, it seems there may be acceptable restrictions in place that prevent the public sharing of your minimal data. However, in line with our goal of ensuring long-term data availability to all interested researchers, PLOS’ Data Policy states that authors cannot be the sole named individuals responsible for ensuring data access.

Response: Thank you. We have revised our data availability statement as per PLOS’ Data Policy.

Response: Thank you so much. All authors, especially the first two authors re-edited the manuscript to remove the grammatical and spelling errors and improve the language use. Besides, we sent the final version for copyedit to a native English speaker, and we trust that she has met the expectations the journal set forth.

Copyedit done by: Sheena Currie, Affiliated with Johns Hopkins University.

Scopus ID: 22633838500

Response to Reviewers’ comments

Response to Reviewer 1 comments

1. Conclusion: L50 talks about the improvement, however the earlier parts don’t talk about assessing this.

Response: Thank you so much. We have revised according to the findings of the study (Please see Line 48-50).

Introduction:

1. L73-6: How are ANC cost effective in current settings? Need elaboration.

Response: Thank you so much for noticing this. We agree that our sentence was a bit over emphasized. We have revised the sentence and added details with relevant citations (Please see Line 74-79).

2. L83 and L86 already answer the objective of the study in Introduction.

Response: Thank you for your meticulous observation. Previous studies focusing on ANC services were mainly limited to a specific region in Afghanistan. However, a national and recent representation is missing in literature. We have added these details into the manuscript (Please see Lines 88 & 92).

Methods:

1. Methods described aren’t the ones opted by the author. They were done as part of the National Survey. These methods should be supported be relevant citations.

Response: Thank you so much. We have added relevant citations (Please see Lines 104).

2. Should also talk about the reasons that were found from women on not opting for ANC. was that a qualitative part of the study? Or reasons were asked and later coded?

Response: Thank you so much for this comment. The reasons for not using the ANC services were collected via a structured questionnaire (the question had several pre-prepared options, each covering one of the reasons we reported in this paper); therefore, it was not a qualitative part of the study. In relation to this and descriptive analysis, we added information in the methods (Please see Lines 135-136).

Results:

1. Table 1 should have p values in a separate column, especially when the result section talks so much about the SIGNIFICANT differences among the variables for the ANC and No ANC utilization. The Asterisks used aren’t very clear because a few variables don’t have any asterisks.

Response: Thank you. We reproduced the table according to your advice (Please see Table 1).

2.There is a lot of repetition in the text and tables.

Response: Thank you so much. We have revised the related text in the manuscript (Please see Lines 155-171).

3. Table 2 can highlight the factors which came significant in final adjusted model so it is easy for reader.

Response: Thanks for this suggestion, we have now highlighted and left only factors that were significant in the final adjusted model (Please see Table 2).

Discussion:

1. Authors can talk more about the modifiable factors that can increase the ANC visits; many factors identified by women who didn’t go for any ANC visits should be worked on.

Response: Thank you for this suggestion. We have incorporated your feedback into the manuscript (Please see Lines 237-242).

Limitations:

1. Please cite where required. Like WHO guidelines L362, please recheck if it is >8 or 4 ANC visits.

Response: Thank you for noticing this. We added a citation to this sentence and checked the whole manuscript for any missing citations. The WHO 2016 ANC guideline recommends a minimum of eight visits for a positive pregnancy experience.

Response to Reviewer 2 comments

I would like to acknowledge the authors of “Predictors of Antenatal Care Services Utilization by Pregnant Women in Afghanistan: Evidence from the Afghanistan Health Survey 2018” for all the hard work on this relevant topic and although the data used from 2018 might not portray the true picture in current scenario but nonetheless would help raise a highly pertinent issue. The authors have done a good job in the analysis and write-up. I have a few suggestions below:

Response: Thank you so much for your constructive feedback. We have incorporated all your feedback in the revised version of our manuscript.

Methods

1. The data analysis part would need more details and also what cutoff was used at the univariate stage to include variables for MV analysis or were all variables included in the MV analysis. Ideally there is a cut-off for variables to be included in the MV.

Response: Thank you for noticing this. Only variables with p value of less than 0.25 in the bivariate analysis were included for multivariate logistic regression analysis. We added this information to manuscript (Please see lines 138-141). 

2. It would have been good if the authors had looked at the factors for early vs. late ANC and if they vary by age on women. I understand this would take time but a suggestion to make this paper more policy relevant.

Response: Thank you for this suggestion. A few authors from this manuscript have already published two papers on the predictors of ANC timing and contents of ANC services, utilizing data from the AHS2018. We have also added citations of these papers to the manuscript for reader’s reference (Please see line 320-326 ).

Papers published already on this topic in Afghanistan:

1. Tawfiq, E., Fazli, M. R., Wasiq, A. W., Stanikzai, M. H., Mansouri, A., & Saeedzai, S. A. (2023). Sociodemographic Predictors of Initiating Antenatal Care Visits by Pregnant Women During First Trimester of Pregnancy: Findings from the Afghanistan Health Survey 2018. International journal of women's health, 15, 475–485. 

2. Stanikzai MH, Tawfiq E, Jafari M, et al. Contents of antenatal care services in Afghanistan: findings from the national health survey 2018. BMC Public Health. 2023;23(1):2469. Published 2023 Dec 11. doi:10.1186/s12889-023-17411-y

Results

1. The length of the results section needs to be shortened especially the description in text for table 1 and fig 1 and 2. As most of the information is there in the table and figures, the text should only focus on the key findings rather than describing all.

Response: Thank you for this suggestion. We agree with you, and we have reduced the text. We have reduced the text related to Table 1 (please see lines 155-171), and reduced text on figures (please see lines 174-188), and on Table 2 (please see lines 197-216).

2. The language and the flow of the writing can be improved especially when describing the MV analysis results, like this is not clear Women’s and husbands’ decision on birth-place choice was a strong predictor of ANC utilization [OR 0.85 (95% CI: 0.74-0.97) for decision made by in laws, and 0.63 (0.55-0.72) for decision made by others for the women, compared to the decision made by women themselves]. The flow and phrasing needs to be clear.

Response: Thank you. We have revised it now (please see lines 210-212). For Table 2, we retained most of the previous text because this is the main analysis of the paper, and readers may be interested to read through the text even if they may look at Table 2.

Discussion

1. This section can be made more succinct as it is currently lengthy.

Response: Thank you so much for this comment. We agree that some sections were voluminous and repetitive. We have revised most sections and added two new paragraphs on your and first reviewer recommendations.

2. Also discuss reasons of why women having late ANC had greater number of ANC visits.

Response: Thank you so much. We have incorporated your feedback (please see lines 301-307).

Figures

1. Fig 1 can be omitted as fig 2 has the required information.

Response: Thank you. With due respect, we want to keep both figures (1 & 2). Figure 1 shows ANC utilization overall and figure 2 compares ANC utilization in urban and rural areas. Since this is the first comprehensive study on ANC utilization in Afghanistan and both figures can provide relevant information for the readers.

We would like to thank once again the respected editor and esteemed reviewers for their constructive feedback and help, and we hope that the revised version of our manuscript will be deemed suitable for publication in this prestigious journal.

---

## [Decision Letter · Decision Letter 1]

3 Jul 2024

PONE-D-24-05385R1Predictors of Antenatal Care Services Utilization by Pregnant Women in Afghanistan: Evidence from the Afghanistan Health Survey 2018PLOS ONE

Dear Dr. Wongrith,

Thank you for submitting your manuscript to PLOS ONE. After careful consideration, we feel that it has merit but does not fully meet PLOS ONE’s publication criteria as it currently stands. Therefore, we invite you to submit a revised version of the manuscript that addresses the points raised during the review process.

We look forward to receiving your revised manuscript.

Kind regards,

Zahra Hoodbhoy

Academic Editor

PLOS ONE

Journal Requirements:

Additional Editor Comments:

Dear Dr Wongrith

Thank you for addressing the initial comments provided by the reviewers. Based on the revised manuscript, the reviewers have requested that some minor comments still need to be addressed.

Reviewers' comments:

Reviewer's Responses to Questions

**Comments to the Author**

1. If the authors have adequately addressed your comments raised in a previous round of review and you feel that this manuscript is now acceptable for publication, you may indicate that here to bypass the “Comments to the Author” section, enter your conflict of interest statement in the “Confidential to Editor” section, and submit your "Accept" recommendation.

Reviewer #1: All comments have been addressed

Reviewer #2: All comments have been addressed

2. Is the manuscript technically sound, and do the data support the conclusions?

Reviewer #1: Yes

Reviewer #2: Yes

3. Has the statistical analysis been performed appropriately and rigorously? 

Reviewer #1: Yes

Reviewer #2: Yes

4. Have the authors made all data underlying the findings in their manuscript fully available?

Reviewer #1: Yes

Reviewer #2: No

5. Is the manuscript presented in an intelligible fashion and written in standard English?

Reviewer #1: Yes

Reviewer #2: No

6. Review Comments to the Author

Reviewer #1: In Abstract Results: Last para needs to be restructured.

L95: I think there's a typo ANC1. please recheck

Figure 1 and 2, can be combined in 1. Please weigh out the importance of keeping figure 3. Figure 4 can better be visualised as a pie chart.

L223: I think the word "only should be removed before 63.2%

Authors must read carefully to make corrections for grammatical errors and sentence structuring.

Reviewer #2: I would like to thank the authors for their revisions and the manuscript looks better, but I would suggest a few pending issues which must be addressed before this manuscript can be considered for publication.

- The language would still need some editing especially in the introduction and the discussion section and please avoid using terms like 'not surprisingly', 'important issue', 'pronounced', 'for instance'.

- Figure 1 and figure 2 should be merged.

- The discussion section is still lengthy and should be shortened.

7. PLOS authors have the option to publish the peer review history of their article (what does this mean?). If published, this will include your full peer review and any attached files.

Reviewer #1: No

Reviewer #2: No

---

## [Author Response · Author response to Decision Letter 1]

13 Jul 2024

We have uploaded a response letter. 

Thank you,

---

## [Editor Report · Decision Letter 2]

9 Aug 2024

Predictors of Antenatal Care Services Utilization by Pregnant Women in Afghanistan: Evidence from the Afghanistan Health Survey 2018

PONE-D-24-05385R2

Dear Dr. Wongwrith,

We’re pleased to inform you that your manuscript has been judged scientifically suitable for publication and will be formally accepted for publication once it meets all outstanding technical requirements.

Kind regards,

Zahra Hoodbhoy

Academic Editor

PLOS ONE

---

## [Editor Report · Acceptance letter]

14 Aug 2024

PONE-D-24-05385R2 

PLOS ONE

Dear Dr. Wongrith, 

I'm pleased to inform you that your manuscript has been deemed suitable for publication in PLOS ONE. Congratulations! Your manuscript is now being handed over to our production team.

Kind regards, 

on behalf of

Dr. Zahra Hoodbhoy 

Academic Editor

PLOS ONE